# Controlling solid-liquid interfacial energy anisotropy through the isotropic liquid

Lei Wang [1,2]*, Jeffrey J. Hoyt[3], Nan Wang [2], Nikolas Provatas[4] & Chad W. Sinclair[1]

Although the anisotropy of the solid-liquid interfacial free energy for most alloy systems is very small, it plays a crucial role in the growth rate, morphology and crystallographic growth direction of dendrites. Previous work posited a dendrite orientation transition via compositional additions. In this work we examine experimentally the change in dendrite growth behaviour in the Al-Sm (Samarium) system as a function of solute concentration and study its interfacial properties using molecular dynamics simulations. We observe a dendrite growth direction which changes from $\langle 100 \rangle$ to $\langle 110 \rangle$ as Sm content increases. The observed change in dendrite orientation is consistent with the simulation results for the variation of the interfacial free energy anisotropy and thus provides definitive confirmation of a conjecture in previous works. In addition, our results provide physical insight into the atomic structural origin of the concentration dependent anisotropy, and deepen our fundamental understanding of solid-liquid interfaces in binary alloys.

[1] Department of Materials Engineering, The University of British Columbia, 309-6350 Stores Road, Vancouver, BC V6T 1Z4, Canada. [2] Key Laboratory of Space Applied Physics and Chemistry, Ministry of Education, School of Science, Northwestern Polytechnical University, 710072 Xi'an, China. [3] Department of Materials Science and Engineering, McMaster University, Hamilton, ON L8S-4L7, Canada. [4] Department of Physics and Centre for the Physics of Materials, McGill University, Montreal, QC H3A 2T8, Canada. *email: lei.wang@mpie.de

Though interfaces constitute a vanishingly small volume fraction of bulk materials, they play an essential role in determining bulk properties. Among the myriad of ways that interfaces impact on properties, one of the most important is their use to control microstructures resulting from phase transitions. In solidification, it is the intrinsic properties of the solid–liquid interface that determines the morphology of the selected product phase and the composition distribution. The interfacial free energy also determines the characteristic scale and morphology of the microstructure of the solid. This connection arises directly from the tendency for pattern formation via dendritic growth[1,2] owing to the solid dendrite arms wanting to grow parallel to crystallographic directions whose interfacial energy is highest. The practical significance of such dendritic growth and its consequences for the properties of bulk products are broad, from the strength of additively manufactured parts[3] to the performance of metal-ion batteries[4]. The significance of this has driven an enormous body of work in the past 10–15 years aimed at identifying the key structural parameters that correlate to the properties of interfaces at the atomistic scale[5].

According to the generally accepted microscopic solvability theory[6,7], dendritic growth velocity and tip radius are sensitively controlled by the (small) anisotropy in the solid–liquid interfacial free energy, $\gamma$. Moreover, the anisotropy also determines the crystallographic growth direction of the dendrites. Pure metals of cubic symmetry will exhibit dendrite arms extending along the $\langle 100 \rangle$ directions so as to minimize the content of high energy (100) surfaces. Due to its importance, experimental and modelling techniques have been devised to estimate the anisotropy of $\gamma$ in several systems[8], yet the atomistic origin of this anisotropy remains elusive[9–11].

With well-controlled experiments and simulations, Haxhimali et al.[12] show that the dendrite growth direction in Al–Zn (zinc) alloys changes with an increase in the solute content. Starting from normal $\langle 100 \rangle$ dendrites in Al-rich alloys, a transition to hyper-branched dendrites and further to $\langle 110 \rangle$ dendrites happens when the content of Zn is increased. The hyper-branched structure is characterized by no preferred growth direction while $\langle 110 \rangle$ dendrites have distinctly crystallographic morphologies but with arms oriented parallel to $\langle 110 \rangle$. The authors showed that the observed dendrite orientation transition (DOT) could be explained by assuming that the solid–liquid anisotropy changes in some fashion with Zn concentration.

Two parameters are often used to characterize the anisotropy of $\gamma$ when the solid phase has underlying cubic symmetry. The first parameter ($\epsilon_1$) represents a fourfold symmetric contribution, and the second ($\epsilon_2$) a sixfold contribution. Using phase-field simulations, Haxhimali et al. found that the space spanned by various values of ($\epsilon_1, -\epsilon_2$) can be separated into three regions where $\langle 100 \rangle$, hyper-branched, or $\langle 110 \rangle$ dendrites dominate the microstructure. Compiling existing estimates of $\epsilon_1$ and $\epsilon_2$ for a variety of pure cubic metals, Haxhimali et al.[12] found that most lie inside the $\langle 100 \rangle$ region or near the $\langle 100 \rangle$/hyper-branched boundary. This is consistent with the dominance of $\langle 100 \rangle$ dendrites observed experimentally. To explain the DOT in Al–Zn it was speculated that the anisotropy must vary with Zn concentration such that the combination of $\epsilon_1$ and $\epsilon_2$ move from the $\langle 100 \rangle$ to $\langle 110 \rangle$ regime. However, it must be stressed that there is no direct evidence why such an anisotropy variation exists in Al–Zn, nor is there any explanation as to why Al–Zn is unique with respect to this solid–liquid interfacial free energy behaviour.

Subsequent molecular dynamics (MD) simulations revealed that the $\gamma$ anisotropy can indeed vary with composition within the ($\epsilon_1, -\epsilon_2$) space. However, for systems examined so far, such as hard-spheres[13], Lennard-Jones[14] and Cu–Ni[15], there is no example where the system can vary sufficiently to predict a

complete transition from $\langle 100 \rangle$ to $\langle 110 \rangle$, which is in contrast with Al–Zn experiments. This illustrates that the DOT phenomenon and the associated anisotropy variations are quite unusual. Interfacial properties are intrinsically determined by the atomic-scale structure of interfaces. For solid–liquid interfaces, a well-accepted feature is that the liquid extending a few interatomic distances from the the crystal exhibits partial ordering. This interfacial ordering has been investigated extensively (see review[9] and references therein) and are related to different interface-control phenomena including nucleation and glass formation. But even though liquid ordering is generally observed, the DOT has only been reported in a few systems. This indicates that the DOT must rely on yet undescribed features of interfaces.

In this work, we demonstrate from laser melting experiments on the Al–samarium (Sm) system that a complete DOT takes place with increasing concentration of Sm. Through MD simulations, we verify that its anisotropy behaviour is consistent with the Haxhimali et al.[12] conjecture of the observed dendrite morphology. We also provide evidence of an unusual atomistic structure of the solid–liquid interface that may provide insights to the origin of the DOT.

## Results

**DOT in Al–Sm alloys.** Figure 1 presents two abnormal microstructures observed in laser melted Al–Sm (samarium) alloys. Figure 1a shows a columnar seaweed microstructure obtained from a laser remelted Al–2.2 at.% Sm alloy. The key feature of a seaweed microstructure is that it exhibits no shape-preserving steady-state tip morphology[16,17]. Such a seaweed microstructure is similar to the hyper-branched structure discussed by Haxihimali et al.[12] and Dantzig et al.[18]. Figure 1b shows equiaxed dendrites obtained from a laser remelted sample of Al–5.6 at.% Sm. The equiaxed dendrites shown in Fig. 1b notably display six

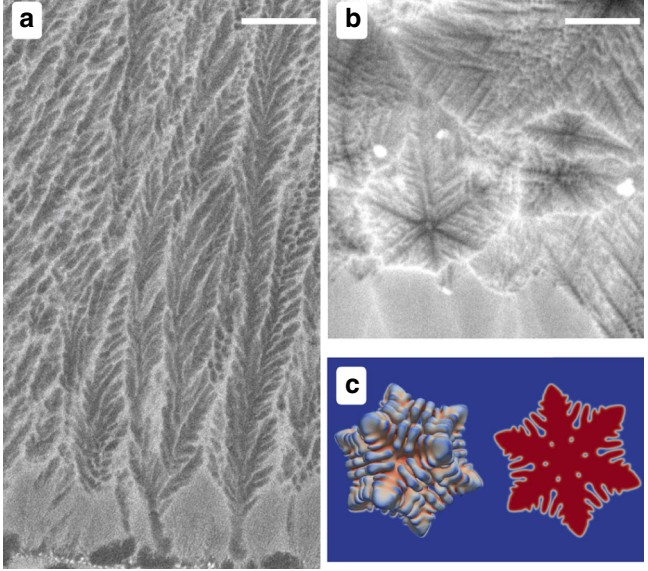

**Fig. 1 Dendrite orientation transition in Al–Sm alloys. a** A columnar dendritic microstructure obtained in a laser remelted Al–2.2 at.% Sm alloy. The microstructure shown here is consistent with a seaweed structure, closely related to a hyper-branched dendritic structure described by Haxihimali et al.[12]. The scale bar indicates 10 μm. **b** Equiaxed $\langle 110 \rangle$ dendrites observed in a laser remelted Al–5.6 at.% Sm alloy (scale bar 2 μm). **c** A three-dimensional dendrite from a phase-field simulation, with a section showing the highlighted six $\langle 110 \rangle$ branches. The microstructure in (**b**) is composed of $\langle 110 \rangle$ dendrites is verified by the observed sixfold symmetry.

branches in one section, implying that these cannot be $\langle 100 \rangle$ dendrites since a $\langle 100 \rangle$ dendrite can have six secondary arms but those can be only simultaneously viewed in three-dimensions (3D), for example, see Fig. 2 in ref. [12]. Six branches in one plane can however be observed with $\langle 110 \rangle$ dendrites. As a comparison, a three-dimensional $\langle 110 \rangle$ dendrite calculated via phase-field simulation is shown in Fig. 1c to illustrate how six branches can lie in one section. Since dendrites grow from pure Al melts are $\langle 100 \rangle$ oriented, Fig. 1 provides compelling evidence of a transition with increasing Sm content in the liquid, from $\langle 100 \rangle$ to a hyper-branched/seaweed structure at approximately 2.2 at.% Sm, and to $\langle 110 \rangle$ at ~5.6 at.% Sm. To make this interpretation fully quantitative, modern high-resolution diffraction techniques could be employed in future studies. It should also be mentioned that the small size of the observed dendrites is due to the small excluded volume available. This may also limit the time/space available for the final arm-selection mechanism to operate. While morphologies can be influenced when dendrites are confined in space/time, quantitative phase-field simulations do not provide evidence of a significant change in the number or orientation of dendrite side-arms once the initial crystallographic pattern emerges.

We attribute the above DOT to solute-induced changes in interfacial free energy anisotropy rather than to possible solute effects on the interfacial kinetics. Although laser melting is often taken as an example of rapid solidification, the solidification conditions vary with position inside the melt pool. The local solidification velocity, $v_n$, is related to the laser beam scanning velocity, $v_B$, by $v_n = \mathbf{v}_B \cdot \mathbf{n}$ where $\mathbf{n}$ is the normal vector to the melt pool boundary after the remelting process reached the steady state. From the bottom of the melt pool $v_n$ increases from zero to the possible order of $v_B$ at the top surface, depending on the specific shape[19]. In our case, the seaweeds and $\langle 110 \rangle$ dendrites are observed near the bottom of melt pools. For the seaweed microstructure, we can estimate the growth velocity to be roughly the order of mm s$^{-1}$ since $v_B = 10$ mm s$^{-1}$. For equiaxed dendrites, the controlling parameter is the undercooling which can be estimated using the columnar to equiaxed transition (CET) model[20]. Specifically, the equiaxed grains under constrained growth conditions form by heterogeneous nucleation driven by the constitutional undercooling ahead of the columnar grains. The undercooling of columnar fronts can be approximated as the bath undercooling of equiaxed grains. Unfortunately, the equiaxed grains near the melt pool boundary as in our case (also observed in ref. [21]) cannot be explained satisfactorily by CET theory since the local solidification velocity and thus undercooling approaches zero there. We believe those equiaxed grains are due to nucleation of pre-existing inoculants with very high potency, perhaps from the first scan. Under these speculated conditions, even a negligible undercooling should be enough to trigger nucleation and growth near the melt pool bottom/boundary although it is difficult for us to put an accurate value on this undercooling based on available data.

At first sight, a direct comparison between experiments under constrained growth conditions and phase-field simulations in free growth conditions is not obvious. We emphasize that as long as kinetic effects on anisotropy are not too strong the transition is expected to happen whether the growth condition is constrained or not. That is the same strategy used by Haxhimali et al.[12]. The main purpose of phase-field simulations is to illustrate how dendritic growth direction changes with the anisotropy and to illustrate six side-arms in 2D is only possible for $\langle 110 \rangle$ dendrites. A quantitative comparison between simulations and experiments in terms of supersaturation and the operating state of dendrite is not attempted here. From previous work[22,23], it is expected that at a speed of ~mm s$^{-1}$, kinetic effects (although present) are not

expected to change the selection mechanisms of dendritic arms for atomically rough interfaces. If the growth rates become sufficiently high such that the interface deviates from local equilibrium, then one needs to consider other factors (including latent heat production and dissipation and the kinetic effect) into a discussion. For the purposes of the following, we deal only with conditions where the local equilibrium approximation is appropriate such that the anisotropy of the interfacial free energy contributes to the dendrite arm orientation primarily.

**Solute-dependent interfacial anisotropy.** To better understand the atomistic origins of the DOT in Al–Sm alloys, we have employed MD simulations. An empirical Finnis-Sinclair type potential developed by Mendelev et al.[24], obtained by fitting the ab initio calculated properties of a range of Al–Sm compounds with compositions close to $Al_{90}Sm_{10}$, was used here. Notably this potential included fitting of the structure of liquid $Al_{90}Sm_{10}$ in its construction. The Al-rich equilibrium phase diagram predicted by this potential for equilibrium between the liquid and Al-rich FCC phase was computed using semi-grand canonical Monte Carlo simulations (see refs. [25,26] and the Methodology section for details of the method employed). A comparison between the CALPHAD (CALculation of PHAse Diagrams) computed[27] Al–Sm phase diagram and that obtained from the Mendelev interatomic potential is shown in Fig. 2. As can be seen, the agreement between the two estimations are good, though the phase diagram obtained from the Mendelev potential exhibits a lower melting point for pure Al and thus a lower liquidus line. Most notable for our purposes is that the Mendelev potential reproduces the negligible solubility of Sm in FCC-Al solid solution.

From the results presented in Fig. 2 it was possible to construct equilibrium solid–liquid interfaces at various temperatures/alloy compositions in MD simulations. Equilibrated solid–liquid interfaces were obtained with (100) and (110) crystal orientations. Once equilibrated, the capillary fluctuation method (CFM)[28,29] was used to obtain the interfacial stiffness, from which $\epsilon_1$ and $\epsilon_2$ can be calculated (see Methods)[12]. Figure 3 illustrates the results of calculated $(\epsilon_1, -\epsilon_2)$ values made for interfaces held at temperatures between 800 and 900 K in 25 K increments. Overlaid on this plot are the expected regimes where $\langle 100 \rangle$,

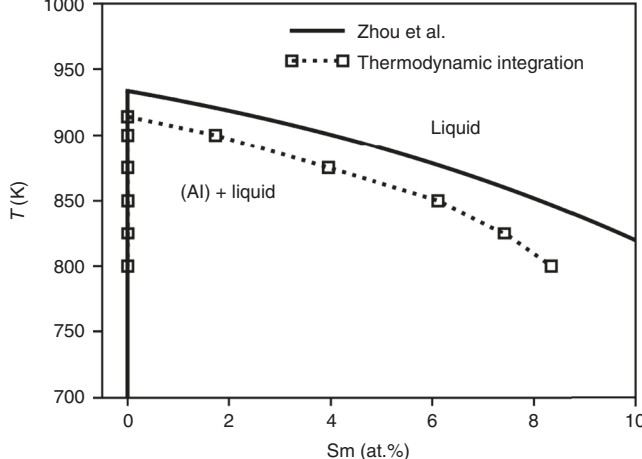

**Fig. 2 Al–Sm phase diagram.** The phase boundaries between liquid and FCC-Al are calculated from semi-grand canonical MC simulations (squares) using the Mendelev interatomic potential[24] and compared to the CALPHAD assessment of Zhou et al.[27] (solid lines). Note that both assessments report ≤100 ppm solubility of Sm in FCC-Al.

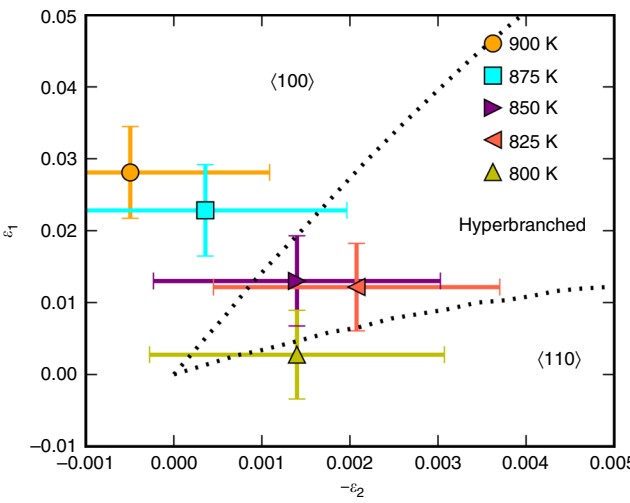

**Fig. 3 Composition-dependent anisotropy parameters $\epsilon_1$ and $\epsilon_2$.** The boundaries (dashed lines) separating different dendritic growth regimes are reproduced from ref. [12]. The coloured points indicate the specific temperatures at which equilibrium MD simulations were performed. As the temperature drops, the Sm composition in the liquid phase increases along the liquidus line while the solid phase remains pure Al as shown by the phase diagram in Fig. 2. The liquid composition increases from ~1.7 at.% at 900 K to ~3.9 at.% (875 K) to ~6.1 at.% (850 K) to ~7.4 at.% (825 K) to ~8.3 at.% at 800 K. As the Sm content in the liquid increases (temperature drops), the interfacial anisotropy changes such that the expected dendritic growth mode shifts from ⟨100⟩ to hyper-branched to ⟨110⟩. The error bars are calculated based on propagation of stiffness uncertainty, as described in Methods.

hyper-branched and ⟨110⟩ oriented dendrites would be found[12]. Within the uncertainty of the calculated values of $\epsilon_1$ and $\epsilon_2$ one can see a clear variation in anisotropy predicted for the Al–Sm alloys, namely ⟨100⟩ oriented dendrites predicted for low Sm contents (<4 at.% Sm, $T > 875$ K), ⟨110⟩ oriented dendrites for high Sm contents (>8 at.% Sm, $T < 800$ K) and hyper-branched microstructures predicted in between.

Both our experiments and atomistic simulations show the full transition from ⟨100⟩ to ⟨110⟩ in Al–Sm alloys. Therefore, the above results represent the definitive confirmation of the Haxhimali et al. DOT conjecture[12]. As discussed above, several MD simulations on various binary alloy systems have shown a variation in anisotropy within the ($\epsilon_1$, $-\epsilon_2$) space, but no previous case has explicitly demonstrated a change from ⟨100⟩ oriented dendrites to the ⟨110⟩ oriented dendrites. Ideally, one would like to compare with experimentally measured anisotropy parameters. In principle, ($\epsilon_1$, $-\epsilon_2$) can be derived from the Wulff shape by measuring the shape of a liquid droplet embedded in a solid matrix. As the anisotropies of solid–liquid interfaces are extremely weak, experimental attempts have been only very limited[30,31]. In addition, the results either suffer from experimental uncertainty or are not consistent with the observed dendrite growth directions.

**Atomic structures of interfaces.** While the above results give clear evidence that changing the Sm content of the liquid can change the anisotropy of the interfacial free energy, the mechanistic origins of this change remain unclear. Another way to view the data shown in Fig. 3 is directly in terms of the difference in interfacial free energy between the ⟨100⟩ and ⟨110⟩ directions (Fig. 4). As can be seen, the anisotropy strength (left axis) decreases while the average interfacial energy (right ordinate) increases with Sm content in the liquid. Both of these trends

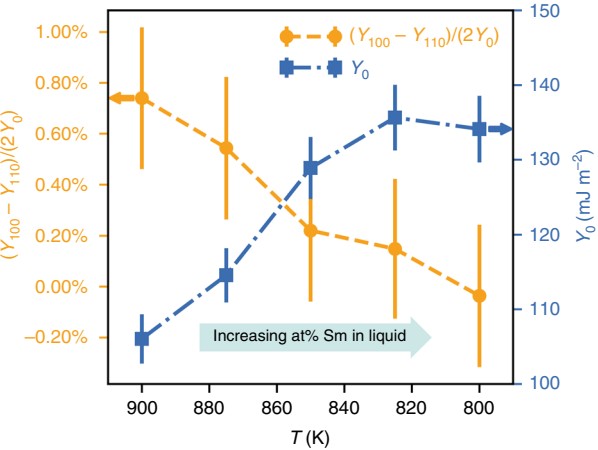

**Fig. 4 Composition-dependent anisotropy strength and interfacial free energy.** The data used are the same as those in Fig. 3. When plotted this way, it shows that the effect of Sm is to drastically reduce the anisotropy strength, meaning that the preference for ⟨110⟩ vs ⟨100⟩ dendrites discussed in relation to Fig. 3 is small. The error bars are calculated based on propagation of stiffness uncertainty.

are unusual in alloy systems and the increase of $\gamma_0$ with decreasing temperature runs counter to the well known result of Spaepen valid for pure metals[10]. However, it is worth mentioning that in a recent MD study by Hoyt et al.[32], the results shown in Fig. 4 are also observed in Cu–Zr alloys. It is also noteworthy that both Al–Sm and Cu–Zr exhibit a very small solid solubility and a large size mismatch. Although it is unclear how these factors directly affect the interfacial energy and its anisotropy, we discuss below how the presence of Sm in Al–Sm alloys creates an unusual solid–liquid interfacial structure, which may provide a possible atomistic origin of the DOT.

Figure 5a shows the fine grained number density as a function of distance perpendicular to the liquid–solid interface for both the (100) and (110) interfaces at the highest (900 K) and lowest (800 K) temperatures studied. Both Al atomic density and the atomic density of the Sm (orange) are shown in the plots. The density profile of Sm is multiplied by a factor of 10 to more clearly compare with the Al density. Focusing on the Al density we see the expected behaviour, with partial ordering extending 3–4 layers into the liquid with a characteristic spacing between peaks similar to that of the solid (see e.g. ref. [13]). However, a striking feature of the density profiles in Fig. 5a is the peaks of the Sm profile do not align with the Al density peaks. To the best of our knowledge this misalignment differs from any previous MD study of the structure of solid–liquid interfaces in other systems, see e.g., Cu–Ni[15] and Lennard-Jones[14]. A solute density profile that is displaced from the total density (dominated by Al) will result in excess number of solute atoms and excess interfacial energy contributions that are not found in the more commonly observed aligned structure and these additional excess quantities will affect the interfacial free energy[32]. From the point of view of the DOT, however, a more significant observation is the fact that the displacement between profiles appears to be greater for the (110) orientation than for the (100) interface for all temperatures studied. Therefore, this unique atomistic structure of the Al–Sm solid–liquid interface provides an additional source of anisotropy, which may play a crucial role in identifying those alloys likely to undergo a DOT.

A possible source of the temperature dependence of the interfacial energy and anisotropy is provided in Fig. 5b. These panels show the total in-plane ordering obtained by time averaging the atomic density in a layer ~3 °A wide (grey shaded

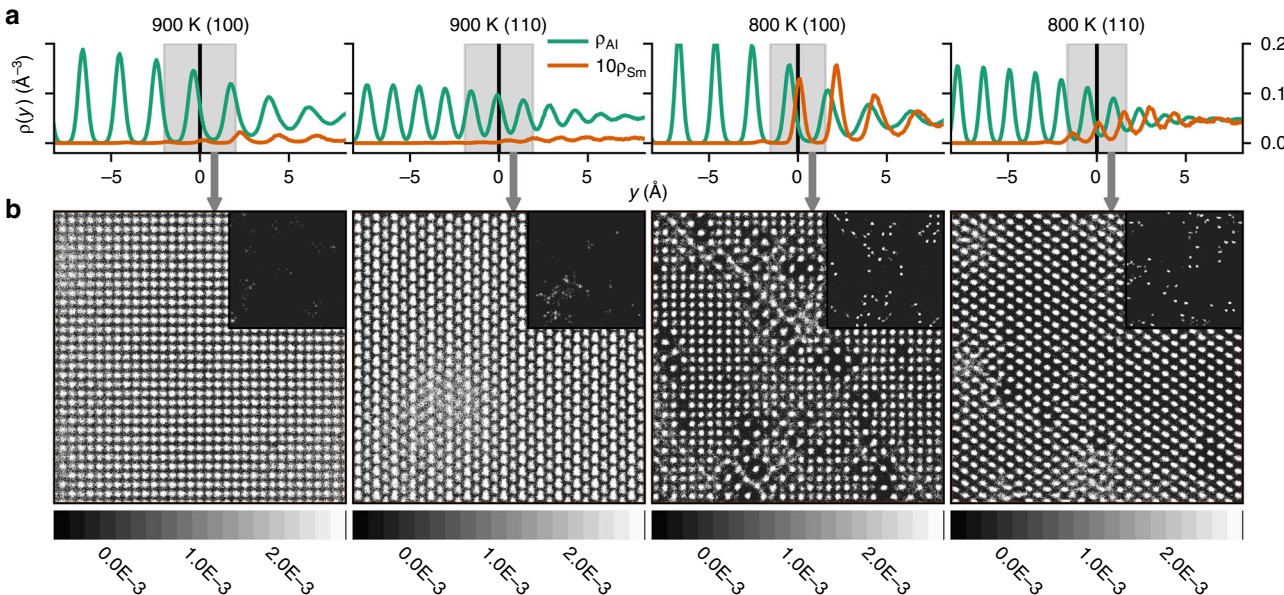

**Fig. 5 Ordering near the solid–liquid interfaces. a** Time-averaged atomic density profiles calculated from layers parallel to the solid–liquid interface for the highest (900 K) and lowest (800 K) temperatures studied. The solid–liquid interface has been set at $y = 0$. The green curve shows the solvent (Al) atomic density while the orange curve shows the average density obtained from Sm only (the Sm density profile is multiplied by 10 so as to plot on the same scale). **b** The total (Al+Sm) in-plane average atomic density. The averages are performed in $y$ direction over the regions identified by the shaded grey boxes in **a** (extending ~3 °A in length perpendicular to the interface). The inset to each figure shows the average atomic density obtained from the Sm atoms alone. The in-plane ordering can illustrate a structure commensurate with the underlying solid. All the profiles are averaged over 500 snapshots (0.5 ns).

box in Fig. 5a centred on the interface position). The insets to Fig. 5b show the corresponding in-plane density for the Sm atoms alone. At the higher temperature (900 K), where the Sm concentration is low, there is little in-plane disordering due to the absence of Sm. On the other hand, at a temperature of 800 K, shown by the right two panels, there is considerable disorder induced by the Sm atoms. The disordering is more pronounced for the (100) interface, a result consistent with the observation discussed above that the two density profiles are more closely aligned for the case of (100). Since the disordered planes will contribute to the excess energy of the interface, the misaligned atomic structure seen in Al–Sm will provide a contribution, not found in most other alloys, to the temperature dependence of $\gamma$ and its anisotropy.

## Discussion

The above observations suggest that in alloys which exhibit an interface structure with misaligned total density (dominated by the solvent density) and the solute density peaks may be candidates for identifying a DOT. It is of course difficult to assess whether the same structure observed in Fig. 5 is also present in the Al–Zn system studied by Haxhimali et al., but it is worth noting that in Al–Zn the equilibrium crystal structure of Zn is HCP and not FCC. Therefore, one can imagine that at high enough Zn content an out-of-phase density structure may be possible, at least in some crystallographic directions. In addition to the structural properties of the solid–liquid interface, the results of this study offer other insights into the DOT and hence the variation in interface anisotropy. Because the crystal induces local ordering in the liquid at the interface, it is tempting to conclude that any variation in anisotropy arises from changes in the crystalline structure with composition (e.g. lattice parameter). But in the case of Al–Sm there is no solubility in the solid and the origin of the anisotropy is necessarily more complicated. Also, while the DOTs in Al–Sm and Al–Zn alloys appear superficially

similar, it must be recalled that Zn, unlike Sm, can be dissolved significantly (up to 67 at.%) into solid Al. This suggests that the solution thermodynamics of the bulk liquid and solid phases do not play a significant role in the explanation of the DOT behaviour.

The results presented above open new challenges and opportunities for furthering our understanding of the role of the liquid–solid interface structure and composition on interfacial free energy and consequently solidification microstructure evolution. If the above interpretation is correct, and alloying elements that disrupt the partial ordering in the liquid at the liquid–solid interface can have a strong effect on both the absolute magnitude and anisotropy of the interfacial free energy, it then suggests that efforts aimed at quantitative prediction of this effect would be desirable. Even without a fully quantitative model, however, this correlation suggests a route to identify alloying elements that would be suitable for inducing a DOT. This in turn would pave the way for additional alloy design possibilities. That is, solute additions whose main role is to alter the dendrite morphology and produce more desirable properties in the solidified material. More broadly, these results also point to the role that solute could play in modifying the partial ordering of the liquid at the liquid–solid interface[33–35]. Aside from changing interfacial free energy anisotropy, this ordering has been cited as an important ingredient for promoting or suppressing crystallization[36,37], this being an important consideration both for grain refinement in commercial alloys and in the production of bulk metallic glasses.

## Methods
**Experiments**. The laser remelting experiments presented in Fig. 1 were performed on material prepared from master alloys prepared as part of a previous study[38]. These alloys were prepared by chill casting 99.999 wt% pure Al and 99.99 wt% Sm in a cylindrical mould water cooled from the bottom. The cast ingots were then cut into plates of thickness of 3 mm. The laser remelting experiments were conducted with a laser operated at 1280 W using a scanning speed of 10 mm s$^{-1}$ for the Al–2.2 at.% Sm (11 wt% Sm) alloys. Due to the formation of large primary intermetallic compounds during chill casting in the Al–5.6 at.% Sm (25 wt% Sm)

alloy, a double scan strategy was used. A first scan at a low speed (21 mm s$^{-1}$) was used to remelt the intermetallic compound and to generate a fine microstructure. The second scan was them performed inside the remelted region of the first scan at a high scan speed (667 mm s$^{-1}$).

**Phase-field simulations.** The quantitative phase-field model presented in Fig. 1 was used to simulate equiaxed dendritic growth[39]. This model has been implemented in a 3D Adaptive mesh refinement Message Passing Interface C++ code as described by Greenwood et al.[40]. The simulations were performed using dimensionless parameters, i.e., $W_0 = \tau = 1$, where $W_0$ and $\tau$ are the characteristic interface width and the kinetic attachment time, respectively. The interfacial free energy anisotropy to produce the results shown were taken as $\epsilon_1 = 0$, $\epsilon_2 = -0.02$. In addition the supersaturation was taken to be $\omega = 0.55$, solute distribution coefficient $k = 0.1$ and dimensionless diffusion coefficient $D = 3$.

**Atomistic simulations.** All atomistic simulations (MD and Monte Carlo simulations) were performed using the LAMMPS (Large-scale Atomic/Molecular Massively Parallel Simulator) software[41]. All calculations were performed using the semi-empirical potential developed by Mendelev et al.[24] for the properties of Al–Sm alloys. The liquidus and solidus boundaries shown in Fig. 2 were obtained using the thermodynamic integration technique described by Ramalingam et al.[42], which computes the relative free energy between solid and liquid phases. The melting point of pure Al which serves as the reference for all free energies is determined first. Starting from the melting point (914 K[24]), the free energy difference between liquid and solid pure Al was found as a function of temperature by a Gibbs–Helmholtz integration. This requires knowledge of the enthalpy per atom of pure Al in the solid and liquid phases at a range of temperatures, which can be determined from MD simulations. The chemical potential difference between Al and Sm in the liquid and FCC phases was then found by performing semi-grand canonical Monte Carlo simulations at the several temperatures between 800 and 900 K with 25 K as the interval. In this case, simulations were performed on boxes containing 6912 atoms in either the liquid or solid FCC state. The chemical potential difference applied to liquid phase ranges from 2.0 to 2.5 eV/atom and 1.9–2.1 eV/atom for solid phase approximately. The systems were equilibrated for a total of 2,360,000 MD time steps and statistics were generated over runs of at least 1,000,000 steps. The equilibrium composition for liquid ranges from 0 to 15 at.% while the solid composition is of the order of 0.1 at.%. The obtained relationship between chemical potential difference and composition was then integrated to obtain the functional form of the Gibbs free energy as a function of Sm content in the liquid and FCC solid phases. From this relationship, the liquidus and solidus lines were determined from a common tangent construction.

To determine the interfacial free energy, the CFM[28,29] has been used. This method provides the interface stiffness, from which interface anisotropy parameters can be computed. For each alloy composition/temperature considered, simulation cells were set up with (100) and (110) oriented crystal-melt interfaces containing 64,000 (20× 20× 40 units cells) and 72,000 atoms (15× 20× 30 unit cells), respectively. For (100) interface, only one stiffness can be obtained due to the fourfold symmetry:

$$\langle A(k) \rangle = \frac{k_{\rm B} T}{aSk^2}, \tag{1}$$

where $A(k)$ is the Fourier amplitude of the interface height profile at a given value of the magnitude of the wave vector $k$, $S$ the stiffness, $a$ the interfacial area, $k_{\rm B}$ the Boltzmann's constant and $T$ absolute temperature. For the (110) interface, the other two stiffnesses can be obtained:

$$\langle A(k_x, k_y) \rangle = \frac{k_{\rm B} T}{a(S_{1\bar{1}0}k_x^2 + S_{001}k_y^2)}. \tag{2}$$

From those three stiffnesses, the interfacial free energy as a function of orientation can be obtained:

$$\frac{\gamma(n)}{\gamma_0} = 1 + \epsilon_1(Q - 3/5) + \epsilon_2(3Q + 66S - 17/7), \tag{3}$$

where $Q = n_x^4 + n_y^4 + n_z^4$ and $S = n_x^2 n_y^2 n_z^2$ are two cubic harmonics. $\gamma_0$ is the average value and $\epsilon_i (i = 1, 2)$ are two adjustable parameters. The error bars are calculated based on the propagation of stiffness uncertainty $\sigma$, which is estimated according to $\sigma^2 = \langle |A(k)|^2 \rangle^2 \tau(k)/t_{\rm run}$ where $t_{\rm run}$ is the total running time and $\tau(k)$ the relax times. $\tau(k)$ can be calculated by integrating time correlation functions $\langle A(k,t)A(-k,0) \rangle^2 / \langle |A(k)|^2 \rangle^2$ derived from the MD data[29]. For each orientation, four replica were used from which to gather statistics. To achieve two-phase equilibrium, the middle half of the system was loaded randomly with the equilibrium concentration of the liquid phase, whereas the two ends of the periodic cell contained the equilibrium concentration of the solid. The middle portion was then melted at high temperatures while the remaining atoms were held fixed. Equilibration was performed for 1 ns in the NP$_y$AT ensemble with $y$ denoting the interface normal direction. During the equilibration runs the solute atoms were periodically swapped with solvent atoms according to a Monte Carlo procedure to achieve different configurations. The frequency dependence of the averaged power

spectrum can then be used to directly obtain the stiffness[28]. In all, 6000 snapshots which last 6 ns were collected for each replica to generate the spectrum. In order to compute the interface position, a local order parameter was defined as $\phi = \frac{1}{12} \sum_i |r_i - r_{\rm fcc}|^2$ where $r_{\rm fcc}$ corresponds to ideal fcc positions in the crystal, $r_i$ denotes the instantaneous atom position, $i$ refers to the nearest neighbour atoms and 12 represents the total number of nearest neighbour atoms over which the summation is taken. The interface location ($y_{\rm loc}$) and width ($W$) were determined by fitting the local order parameter profile across the interface to the hyperbolic tangent function, $c_1 + c_2 \tanh((y - y_{\rm loc})/W)$.

## Data availability
The data that support the findings of this study are available from the corresponding author upon request.

## Code availability
The codes that support the findings of this study are available from the corresponding author upon request.

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

## Acknowledgements

We thank Compute Canada for computing resources. We also thank Mike Greenwood who developed, with N.P., the phase-field code used herein. N.P. would like to thank the Canada Research Chairs (CRC) program and N.P. and C.S. the National Science and Engineering Research Council of Canada (NSERC) for funding.

## Author contributions

The first idea started from discussions about the seaweed formation between L.W. and N.P. at McGill University. L.W. observed the abnormal dendritic morphologies in Al–Sm alloys and initiated the collaboration with the other authors. L.W. and J.J.H calculated the interface energy anisotropy. N.W. did the laser remelting experiments. L.W. and N.P. conducted the phase-field simulations. C.S. contributed his expertise in atomic-scale simulations and interfacial thermodynamics. All authors co-edited the manuscript.

## Competing interests

The authors declare no competing interests.
