## [Peer Review File · Nature Communications]

Reviewers' Comments:

Reviewer #1:

Remarks to the Author:

In the manuscript "Controlling solid-liquid interfacial energy anisotropy through the isotropic liquid", Wang et. al. describe new insight that the presence of an alloying component, if insoluble in the solid phase, can decrease the interface anisotropy and alter the dendritic growth mode in a model system of Al-Sm.

Unfortunately, I do not find the present manuscript to be suitable for publication in Nature Communications. The principle reasons are:

(1) Is this a more general phenomena or something specific to the Al-Sm system? If the authors can comment on this point then it would make a stronger case for the broad audience of Nature vs. a more disciplinary journal.

(2) Once a concentration gradient of the non-soluble component has formed at the interface, it will influence the propagation of latent heat, which obviously plays a key role in the resulting interface instabilities. Yet the authors did not even comment in passing on this rather important point. This is a substantial flaw in the current work and needs to be corrected, even if sent to a more discipline-specific journal.

If the above two critiques can be adequately addressed, then I would be willing to consider the manuscript again for possible publication .

Some more detailed points of revision:

(3) Fig. 2 is not production-ready. The caption needs to be descriptive. One should not just refer to "Thermocalc" (commercial software) but rather specify more precisely the underlying method. The phase diagram should be labelled, e.g. "soluble" and "insoluble", etc.

(4) The captions are highly minimalistic, which is not appropriate for a letter. Fig. 3 and 4 need more details in the captions. Figure should be able to stand on their own without hunting through the text.

(5) The methods section: experimental part is OK but theoretical section is wholly lacking in details. In it's current state it's not clear that a reader would have enough information to reproduce the reported work. For example, "the thermodynamic integration procedure is used..." needs more details (e.g., window size, etc) since the resulting free energies can be quite sensitive to those details; it also need additional citations for the non-expert reader. Likewise the CFM method needs a citations or two (indeed one of the current authors has numerous works in this area so I would cite something appropriate). "The interface location...determined by fitting the local order parameter profile..." but what is the order parameter? Steinhardt Q_l ? It needs to be specified if one is to reproduce these results.

I'm willing to reconsider my position on this manuscript, but only if the above points are clearly addressed

Reviewer #2:

Remarks to the Author:

No comments to the authors.

Reviewer #3:

Remarks to the Author:

Wang et al. performed phase field modeling and atomistic simulations to explain the novel experimental observation that the dendritic growth behavior changes significantly with a minor change of the solute concentrations. Their main finding is that the anisotropy of the solid-liquid interfacial free energy is the dominant factor here, and the anisotropy was changed due to the solid-liquid interface structure instead of the solid phase according to the understanding of prevalent theory. The authors chose a proper system where the effects of solubility are eliminated, so that they have a clear condition of the unchanged solid properties to draw the conclusions. The results of the experiments, mesoscopic modeling and atomistic simulation are coherent and consistent. Before recommending this work for publication in Nature Communications, I would like the authors to address the following issues:

1. The authors should clearly articulate the impact of their results on the scientific community and why the paper is of interest to a broad range of readers.
2. The incompatibility of Sm on fcc-Al interface shown in Fig.5 has been reported before. I think what's more important here is to address how differently Sm changes the ordering on [100] and [110]. This should be the essential origin of the anisotropy in this alloy. Qualitatively, it seems the interface on 800 K (110) maintains more ordered fcc ordering than the one on 800K (100) in Fig. 5b. Authors should provide some quantitative analysis and discussion on it. The resolution of Fig. 5b should also be improved.
3. I suppose the Sm concentrations in Fig.3 are different at each temperature. If so, it should be indicated in the figure caption.

Response to the reviewer 1

We thank the reviewer 1 for the critical assessment of our work. In the following we address his/her concerns point by point.

In the manuscript "Controlling solid-liquid interfacial energy anisotropy through the isotropic liquid", Wang et. al. describe new insight that the presence of an alloying component, if insoluble in the solid phase, can decrease the interface anisotropy and alter the dendritic growth mode in a model system of Al-Sm. Unfortunately, I do not find the present manuscript to be suitable for publication in Nature Communications. The principle reasons are:

(1) Is this a more general phenomena or something specific to the Al-Sm system? If the authors can comment on this point then it would make a stronger case for the broad audience of Nature vs. a more disciplinary journal.

We believe that the consequences of this work are broad, with potential importance to both the fundamental knowledge of the atomistic origins of the interfacial free energy in solid-liquid (and potentially solid-vapor) systems as well as on the practical consequences of dendritic pattern formation and microstructure development in materials. The results for the Al-Sm alloy presented here represent one example of a much broader set of phenomena related to the impact of solute on near-interface liquid ordering that has been debated theoretically for many years. To our knowledge, this is the first attempt to make a link between this ordering and the thermodynamic properties of the solid-liquid interface. This opens new avenues for fundamental exploration as well as the potential for practical tailoring of material properties through judicious selection of alloy compositions. To better address this we have made several changes to the text of the manuscript:

(i) In the introduction we have better tried to situate this work in the broad activities currently under active exploration by the research community related to elucidating the atomistic origins of properties at liquid-solid interfaces.

(ii) To better illustrate that the effects shown here are not unique to the Al-Sm alloy system, we have tried to better relate this work to recent work in other alloy systems. For example, we have related our work to the results from very recent atomistic simulations on Cu-Zr (Computational Materials Science 154 (2018) 303308) showing a similar, though less strong effect.

(iii) Finally, in our conclusion, we have highlighted the potential that exists for the results of this work to be applied to other interface phenomena. We believe, for example, that tailoring the partial liquid ordering through solute additions may be significant for understanding nucleation and glass formation in metallic systems. It is notable that Al-Sm alloys (as well as Cu-Zr referenced above) are both glass formers.

(2) Once a concentration gradient of the non-soluble component has formed at the interface,

it will influence the propagation of latent heat, which obviously plays a key role in the resulting interface instabilities. Yet the authors did not even comment in passing on this rather important point. This is a substantial flaw in the current work and needs to be corrected, even if sent to a more discipline-specific journal.

The present study focuses entirely on the equilibrium thermodynamic properties of the solid-liquid interface. It is true that thermal gradients and latent heat evolution can modify the microstructures developed during solidification at finite rates especially if these rates start to compete with the rate of atomic attachment to the liquid/solid interface. However, as pointed out by Haxhimali *et al.* (Nature Materials 5, 660664 (2006)) for the vast majority of cases of practical interest, the attachment kinetics for atoms at the solid-liquid interface are fast enough relative to the dendrite growth rates that local equilibrium will prevail and that the anisotropy of the interfacial free energy is primarily determines the dendrite growth direction. Actually, in our MD simulations, the interface is fluctuating but not moving. To make it clear that we are limiting ourselves to situations where the dendrite arm orientation is primarily determined by the interfacial free energy anisotropy, we have added the following to page 3 of the manuscript:

If the dendrite growth rates become sufficiently high, such that local equilibrium can not be maintained at the interface, then one would have to consider other factors (including latent heat production and dissipation) into a discussion of dendrite growth. For the purposes of the following, we deal only with conditions where local equilibrium is appropriate such that the anisotropy of the interfacial free energy can be considered as the primary contributor to the dendrite arm orientation.

If the above two critiques can be adequately addressed, then I would be willing to consider the manuscript again for possible publication. Some more detailed points of revision:

(3) Fig. 2 is not production-ready. The caption needs to be descriptive. One should not just refer to "Thermocalc" (commercial software) but rather specify more precisely the underlying method. The phase diagram should be labelled, e.g. "soluble" and "insoluble", etc.

This figure has been modified and the source of the CALPHAD data (Zhou SH, Napolitano R. Metall Mater Trans 2008;39A:502.) used to construct the phase boundary is given rather than the reference to the software that stored it. We have also explicitly labeled the phase domains (Liquid + FCC Al and Liquid) on the diagram.

(4) The captions are highly minimalistic, which is not appropriate for a letter. Fig. 3 and 4 need more details in the captions. Figure should be able to stand on their own without hunting through the text.

We agree with the reviewer and so have re-written all of the figure captions so as to make them more descriptive. We do not reproduce them all here but hope that the reviewer will be able to see the changes directly in the manuscript.

(5) The methods section: experimental part is OK but theoretical section is wholly lacking in details. In its current state it's not clear that a reader would have enough information to reproduce the reported work. For example, "the thermodynamic integration procedure is used..." needs more details (e.g., window size, etc) since the resulting free energies can be quite sensitive to those details; it also needs additional citations for the non-expert reader. Likewise the CFM method needs a citation or two (indeed one of the current authors has numerous works in this area so I would cite something appropriate). "The interface location...determined by fitting the local order parameter profile..." but what is the order parameter? Steinhardt Q_L? It needs to be specified if one is to reproduce these results.

We have re-written the methods section to provide sufficient details for all of the simulation work to be reproduced. This includes details on the system sizes, calculation ensembles, how the interface position was identified (and what order parameter was used). We do not reproduce the entirety of the methodology section here but hope that the reviewer will be able to reference to the changes made in the updated manuscript.

I'm willing to reconsider my position on this manuscript, but only if the above points are clearly addressed.

Response to the review 3

We thank the reviewer 3 for the critical assessment of our work. In the following we address his/her concerns point by point.

Reviewer 3 (Remarks to the Author): Wang et al. performed phase field modeling and atomistic simulations to explain the novel experimental observation that the dendritic growth behavior changes significantly with a minor change of the solute concentrations. Their main finding is that the anisotropy of the solid-liquid interfacial free energy is the dominant factor here, and the anisotropy was changed due to the solid-liquid interface structure instead of the solid phase according to the understanding of prevalent theory. The authors chose a proper system where the effects of solubility are eliminated, so that they have a clear condition of the unchanged solid properties to draw the conclusions. The results of the experiments, mesoscopic modeling and atomistic simulation are coherent and consistent. Before recommending this work for publication in Nature Communications, I would like the authors to address the following issues:

1. The authors should clearly articulate the impact of their results on the scientific community and why the paper is of interest to a broad range of readers.

We believe that the consequences of this work are broad, with potential importance to both the fundamental knowledge of the atomistic origins of the interfacial free energy in solid-liquid (and potentially solid-vapor) systems as well as on the practical consequences of dendritic pattern formation and microstructure development in materials. The results for the Al-Sm alloy presented here represent one example of a much broader set of phenomena related to the impact of solute on near-interface liquid ordering that has been debated theoretically for many years. To our knowledge, this is the first attempt to make a link between this ordering and the thermodynamic properties of the solid-liquid interface. This opens new avenues for fundamental exploration as well as the potential for practical tailoring of material properties through judicious selection of alloy compositions. To better highlight this we have made several changes to the text of the manuscript:

(i) In the introduction we have better tried to situate this work in the broad activities currently under active exploration by the research community related to elucidating the atomistic origins of properties at liquid-solid interfaces.

(ii) To better illustrate that the effects shown here are not unique to the Al-Sm alloy system, we have tried to better relate this work to recent work in other alloy systems. For example, we have related our work to the results from very recent atomistic simulations on Cu-Zr (Computational Materials Science 154 (2018) 303308) showing a similar, though less strong effect.

(iii) Finally, in our conclusion, we have highlighted the potential that exists for the results of this work to be applied to other interface phenomena. We believe, for example, that tailoring the partial liquid ordering through solute additions may be significant for understanding nucleation and glass formation in metallic systems. It is notable that Al-Sm alloys (as well as Cu-Zr referenced above) are both glass formers.

2. The incompatibility of Sm on fcc-Al interface shown in Fig.5 has been reported before. I think what's more important here is to address how differently Sm changes the ordering on [100] and [110]. This should be the essential origin of the anisotropy in this alloy. Qualitatively, it seems the interface on 800 K (110) maintains more ordered fcc ordering than the one on 800K (100) in Fig. 5b. Authors should provide some quantitative analysis and discussion on it. The resolution of Fig. 5b should also be improved.

We agree with the reviewer that our description of the changes to the ordering were not sufficient to clearly illustrate the link between the changed (reduced) anisotropy and the addition of Sm to the liquid. We have re-written the section dealing with our description of the layering and in-plane ordering to make this link

clearer. At this point we do not have a fully quantitative theory to link the changes in liquid partial ordering to the changes in the interfacial free energy but we believe that the observations presented here, along with their interpretation, points to interesting clues and ideas for further exploration, ones that seem to align with other work that we know is ongoing. We hope that with this short note we can provoke excitement for some further exploration between this effect of partial liquid ordering, solute and interfacial properties. The resolution of Fig. 5b has been improved.

3. I suppose the S_m concentrations in Fig.3 are different at each temperature. If so, it should be indicated in the figure caption.

We have updated the figure caption and the figure legend to make the composition of the liquid for each of these simulations clearer.

Response to the Editors Comments

These concerns include, but are not limited to, the generality of the phenomenon observed here (Reviewer 1 and 3), the effect of a concentration gradient on latent heat propagation (Reviewer 1), and how S_m impacts the interfacial ordering (Reviewer 3). Additionally, more detailed clarifications on your experimental and simulation methods, discussion of how S_m affects the local ordering in the liquid, and how the current findings can be placed within the context of A. Hashibon, J. Adler, M. W. Finnis and W. D. Kaplan, Ordering at solid-liquid interfaces between dissimilar materials, *Interface Science*, 9 [3-4] 175-181, 2001, would be appreciated.

We believe that we have dealt with all of the points raised here, including the addition of further references to older work examining liquid partial ordering (including the work of Hashibon and co-workers). In order to do so, the introduction and the discussion/summary have been largely rewritten. We hope that you will find the revisions suitable for this manuscript to be re-reviewed.

Reviewers' Comments:

Reviewer #3:

Remarks to the Author:

With the changes made to the manuscript I recommend the paper to be published in Nature Communications.

Reviewer #4:

Remarks to the Author:

General comments:

The authors report on work that provides a corroborating example to support Haximali's conjecture regarding a general correlation between anisotropy and dendrite orientation selection in low-anisotropy metallic (rough interface) dendrites. This is an important topic, considering the breadth of properties that are ultimately controlled by dendrite tip selection behavior.

For Al-Sm alloys of 5.6 at% Sm, $\langle 110 \rangle$ fcc-Al dendrites were produced using a double-scan laser-trace melting technique and observed by electron microscopy. For more dilute compositions (e.g. 2.2 at% Sm), a seaweed structure was observed. Considering that fcc-Al dendrites generally grow with a $\langle 100 \rangle$ orientation, this observation is taken as an indication of a $\langle 100 \rangle$ to $\langle 110 \rangle$ dendrite orientation transition (DOT), induced by the influence of Sm solute on the anisotropy of interfacial free energy.

To further investigate their assertion, the authors use established MD potentials and apply the CFM method to determine the solid-liquid interfacial free energy for different Al-rich compositions in the Al-Sm system. Specifically, the $\langle 100 \rangle$ and $\langle 110 \rangle$ orientations were computed to assess the crystallographic anisotropy of the interfacial free energy, summarized by the ϵ_1 and ϵ_2 coefficient values (as in ref [12]). The liquid compositions for the MD simulations were taken as the equilibrium liquidus compositions for the simulation temperature used. Plotting the results on the Haximali-type plot indicate that a $\langle 100 \rangle$ to $\langle 110 \rangle$ DOT may be expected with the composition variation examined by CFM (1.7 to 8.3 at% Sm). In addition to providing an example supporting Haximali's conjecture, the results reported here indicate that anisotropy decreases with Sm content, while the interfacial free energy increases in magnitude. The authors discuss this interesting result in terms of liquid-phase ordering near the solid. (Surprisingly, the authors do not cite recent literature on liquid ordering in Al-Sm, including near-interface effects.)

The experimental evidence presented (in Fig. 1) has some limitations that deserve mention. The observation that the transition has occurred is rather weak, at least based on the evidence presented. The orientations are not confirmed by diffraction, only the appearance of six arms in an apparent plane (and a comparison with a phase-field simulation). The observed crystals, presumably exhibiting the $\langle 110 \rangle$ orientation, are very small equiaxed crystals, not necessarily demonstrating any sort of stable growth that was "selected" over any time or distance. Finally, no examples of $\langle 100 \rangle$ dendrites are shown, for comparison.

The laser scan microstructures were observed somewhere "at the very bottom of melt pools." Without a more detailed description of the melt pool shape and location of these observations, the prevailing local growth velocity remains indeterminate. This is particularly important here since this velocity along with the dendrite tip temperature and supersaturation (thus, solid and adjacent liquid compositions) are intimately connected. These could have been computed from the relevant analytical growth model (knowing the composition, local isotherm velocity, and thermal gradient). This problem is further amplified by the double-scan method employed, where the very high scan rate (667 mm/s) is used in the remelting of a previously melted (and potentially segregated) laser trace. If some steps were taken to establish the segregation patterns from the first scan, these

should be stated. In addition, the attribution of the observed orientation transition to solute-induced interfacial free energy effects (rather than solute induced kinetic effects) is difficult to defend without a more explicit assessment of local conditions or some comparison with other locations in the melt pool where velocity-sensitivity would be observed, if present.

The laser scanning method produces a condition of directional solidification (thermally constrained growth). However, the dendritic structures computed by phase-field simulation were generated as equiaxed dendrites (thermally unconstrained) with a specific (perhaps arbitrary) supersaturation. Given this difference, the value of the comparison in Fig. 1 is not clear. Certainly, Figs. 1b and 1c demonstrate that $\langle 110 \rangle$ are plausible, but the difference in conditions does not add any clarity to the question involving the role of solute content, as discussed in previous comment.

Overall assessment:

While the work is important and potentially highly compelling, these shortcoming issues must be addressed before the manuscript can be responsibly published.

Minor points:

It may be a stretch to refer to the 2006 study by Haxhimali as "recent".

The explanation of the seaweed structure (on page 2) implies that successive tip splitting is the defining characteristic of this morphology. The defining feature is that no shape-preserving steady-state tip morphology is achieved (whether one exists or not is another matter).

There are numerous places in the manuscript in need of careful proofreading and correction with regard to the use of English. These detract from the scientific content.

2 Response to the 4th Reviewer's comment on our responses to 1st reviewer's first point

Reviewer 1 raises a fair point here. The main issue is how solute in the near-interface liquid influences atomic ordering in this system, how this order affects interfacial free energy, and whether this is likely to be a general behavior. The authors have addressed this concern in the revised manuscript by revising the manuscript, as summarized by points (i-iii) above. In my opinion, the revision to the introduction (point i) provides little additional perspective relative to this issue. Rather, the authors simply call attention to the well accepted (but not well quantified) link between the anisotropy of interfacial free energy and the selection of growth morphology (e.g. in dendritic growth of atomically rough phases). Moreover, in the revised text, it appears that the authors overstate the directness of this relationship, remaining tacit on the interplay with interface kinetics (also anisotropic) and other dynamics. The concluding remarks (point iii) go a little further to shed some light on the possible broad implications of this behavior, suggesting that solute effects on interface structure may be contributory to the promotion or suppression of nucleation, as might be observed as grain refinement or glass formation, respectively. An introduction that provides this perspective would be more appropriate. (There is a considerable body of reported literature on composition-dependent ordering in metallic liquids. In addition, there are reports of near-interface liquid ordering during solid growth in this particular system, albeit for different phases. e.g. (1, 2)). Finally, I would like to offer my opinion that the authors best response to comment (1) might simply be to call attention to Fig.5 (a,b) and the associated text (two main paragraphs on page 4). Herein lies the main contribution of the paper and the best potential line of connection to broad behavior.

1. Y. Sun et al., *Mater. Lett.* 186, 2629 (2017).
2. Y. Sun et al., *Model.Simul.Mater.Sci.Eng.*26(2018),doi:10.1088/1361-651X/aa9747.

We thank the reviewer 4 for the critical assessment. The introduction part actually changed substantially since the first version as we decided later to make two special points very clear rather than overemphasize the

general liquid ordering. We will first give the two points and then explain the reason for doing so. This first point we made is that Al-Sm is the first case in which DOT transition is observed in experiments and verified by MD simulations. It is also the first system which shows a *full* transition from $\langle 100 \rangle$ to $\langle 110 \rangle$ predicted by MD simulations. The second point is we noticed that the layering of Sm is always shifted relative to the layering of Al, a feature not shown by Sun and colleagues perviously. We agree with the reviewer that a vast literature exist on this ordering near the solid-liquid interface, but not extremely helpful for us here for two reasons. The first one is it is difficult to link this ordering with interfacial properties as the partial-ordered structure is hard to quantify. What is more important is: liquid ordering near the interface is general, but only some systems can exhibit this DOT. This indicates that liquid ordering in general is not the reason for the transition, it must be some unknown ordering features. In discussion, we try to link the mismatch of Sm-layering relative to Al-layering and the anisotropy change. We extend this further by exploring more fundamental parameters which control this special layering mismatch. We suggest that the mismatch of lattice constants between Al and Sm is the possible reason and propose systems which are likely to be candidates for such a transition. In our opinion, this is a fundamental way to address ‘general phenomena and general interest’ question. Again, we stress that the liquid ordering near the interface is general and well-accepted but those two points are actually our contribution. Following the above considerations, we think many points are NOT obvious until Figure 5 and it is therefore more logical to us to put them in discussion rather than in introduction. We however incorporated the reviewer’s suggestion to make the structure more logical.

3 Response to the 4th reviewer’s own comments

1. The experimental evidence presented (in Fig. 1) has some limitations that deserve mention. The observation that the transition has occurred is rather weak, at least based on the evidence presented. The orientations are not confirmed by diffraction, only the appearance of six arms in an apparent plane (and a comparison with a phase-field simulation). The observed crystals, presumably exhibiting the $\langle 110 \rangle$ orientation, are very small equiaxed crystals, not necessarily demonstrating any sort of stable growth that was selected over any time or distance. Finally, no examples of $\langle 100 \rangle$ dendrites are shown, for comparison.

We agree with the reviewer that high-resolution EBSD would be the best choice here to confirm orientation selection, since the grains in our experimental samples are quite small. Moreover, the small size of the dendrites is due to the small excluded volume available to growing dendrites, which may influence the time/space available for the final arm-selection mechanism to take place. We comment here that, based on quantitative phase field simulations, we do not generally observe a significant change in the number or orientation of dendrite side-arms when dendrites are confined in space/time; it is typically the arm length, speed and width that is impacted the most. Moreover, in our samples. we observe multiple grains (albeit less clear than the one shown in Fig. 1) that exhibit strange side-branching direction of dendrite arms. This fact, together with MD simulations, leads us to believe that these structures are exhibiting a transition in dendrite arm direction. Although we do not have experimental results on pure Al, it is well accepted that dendrites in pure Al melt grow along $\langle 100 \rangle$ and the 3D structures of $\langle 100 \rangle$ dendrites with great details can be found in literature, for example papers by Dantzig *et al.* We have revised the manuscript to highlight this important reviewer comment and our above response.

2. The laser scan microstructures were observed somewhere at the very bottom of melt pools. Without a more detailed description of the melt pool shape and location of these observations, the prevailing local growth velocity remains indeterminate. This is particularly important here since this velocity along with the dendrite tip temperature and supersaturation (thus, solid and adjacent liquid compositions) are intimately connected. These could have been computed from the relevant analytical growth model (knowing the composition, local isotherm velocity, and thermal gradient). This problem is further amplified by the double-scan method employed, where

the very high scan rate (667 mm/s) is used in the remelting of a previously melted (and potentially segregated) laser trace. If some steps were taken to establish the segregation patterns from the first scan, these should be stated. In addition, the attribution of the observed orientation transition to solute-induced interfacial free energy effects (rather than solute induced kinetic effects) is difficult to defend without a more explicit assessment of local conditions or some comparison with other locations in the melt pool where velocity-sensitivity would be observed, if present.

The melt pool shape will be the constant for an observer who moves at the same speed as the laser beam when the remelting process reaches the steady state. The local solidification velocity, v_n , can be linked to the beam scanning velocity, v_B , by $v_n = \vec{v}_B \cdot \vec{n}$ where \vec{n} is the normal vector to the melt pool boundary. For dendrites that do not follow the heat-flux direction, the tip growth velocity can also be determined if the tip growth direction, relative to \vec{n} is known. The details can be found in several papers by W. Kurz and colleagues, (e.g. Acta mater. 49 (2001) 1051 and Acta mater. 37(1989) 3305). Based on this scheme, we can estimate the growth velocity for columnar seaweed to be a few mm/s since $v_B = 10\text{mm/s}$. While present, kinetic effects responsible for non-equilibrium solute partitioning are not expected to change the morphological selection mechanisms of dendritic forms at these velocities for atomically rough interfaces (T. Pinomaa and N. Provatas, Acta Materialia, Vol. 168, 167 (2019)).

For the case of $\langle 110 \rangle$ equiaxed dendrites, the undercooling is the controlling parameter to be estimated, which can be achieved using the columnar to equiaxed transition (CET) model. To be more specific, the formation of equiaxed grains under directional solidification conditions can often be explained by heterogeneous nucleation driven by the constitutional undercooling ahead of the columnar grains. The columnar tip undercooling can be taken as the bath undercooling of equiaxed grains, as an approximation. Unfortunately, it turns out that the CET model cannot explain our case. The reason is that the melt pool boundary is not prone to nucleation since the local solidification velocity and thus undercooling approaches zero there. But what we observed is equiaxed grain formation just near the melt pool boundary, a phenomenon also observed in laser remelted Ni-based superalloys (Phd Thesis by S. Mokadem, EPFL, 2004). We therefore believe those equiaxed grains are due to the nucleation of pre-existing inoculants with very high potency, perhaps from the first scan. In such case, nucleation and growth new grains can be triggered by a very small undercooling. Under this speculated conditions, $\langle 110 \rangle$ equiaxed dendrites at the melt pool bottom/boundary are not expected to have a large bath undercooling although it is difficult for us to put an accurate value based to the available data. As for the segregation during the double scan, since the second scan was conducted immediately after the sample cooled down from the first scan, the segregation is not quantified. However, compared to the cast microstructures, the first scan is expected to give finer microstructures with less segregation, which serve as a better matrix for the second scan.

3. The laser scanning method produces a condition of directional solidification (thermally constrained growth). However, the dendritic structures computed by phase-field simulation were generated as equiaxed dendrites (thermally unconstrained) with a specific (perhaps arbitrary) supersaturation. Given this difference, the value of the comparison in Fig. 1 is not clear. Certainly, Figs. 1b and 1c demonstrate that $\langle 110 \rangle$ are plausible, but the difference in conditions does not add any clarity to the question involving the role of solute content, as discussed in previous comment.

We agree with the reviewer that many aspects can be different for solidification in constrained and non-constrained growth conditions (see for example section 4.1 in book Fundamentals of solidification), however that is not the case for dendrite orientation transition, which is controlled by interfacial anisotropy. In our case, we take the anisotropy of interfacial free energy as the dominant effect. As long as this anisotropy can change from $\langle 100 \rangle$ to $\langle 110 \rangle$ in the $(\epsilon_1, -\epsilon_2)$ space, this transition will happen no matter the solidification condition is constrained or not, *as long as the interfacial kinetics anisotropy is not significant*. This is exactly reason why the orientation transition reported by Haxhimali *et al.* is also observed in constrained growth conditions while their phase-field modelling was done with uniform supersaturation. Actually, from the work of Bragard *et al.* (Bragard *et al.* Interface Science (2002) 10: 121.), it is expected that kinetic effect,

although may still present, will not change the transition at the speed of $\sim\text{mm/s}$ relevant to our experiments. Note that we do not attempt to make a quantitative comparison between phase-field and experiments in those sub-figures. Phase-field simulations can clearly illustrate how dendritic growth direction changes with anisotropy as shown previously by Haxhimali *et al.* For us, the aim is to demonstrate that how six arms in 2D is possible for $\langle 110 \rangle$ dendrite while this is not possible for $\langle 100 \rangle$ dendrites. In conclusion, we expect the changes in orientation supported by the data of Figure 1 is valuable *as long as kinetic effects on anisotropy are not too strong*.

Overall assessment: While the work is important and potentially highly compelling, these shortcoming issues must be addressed before the manuscript can be responsibly published.

4. Minor points:

- (a) It may be a stretch to refer to the 2006 study by Haxhimali as recent.
- (b) The explanation of the seaweed structure (on page 2) implies that successive tip splitting is the defining characteristic of this morphology. The defining feature is that no shape-preserving steady-state tip morphology is achieved (whether one exists or not is another matter).
- (c) There are numerous places in the manuscript in need of careful proofreading and correction with regard to the use of English. These detract from the scientific content.

We altered the manuscript to address the first two comments. We have also re-edited the manuscript to catch any English grammar issues.

Reviewers' Comments:

Reviewer #3:

Remarks to the Author:

I believe the response is sufficient to address the comments of Reviewer 4.

It is a long-standing issue whether and how near-interface order affects the interfacial free energy.

The authors made a good argument in this paper that interfacial ordering is general while dendrite orientation transition is not. They attributed the key factor to the mismatch of Sm and Al layering in the current case. After the long review, the current manuscript presents a consistent combination of

experiments, phase field and atomic simulations. In my opinion, the current manuscript is acceptable

for publication in Nature Communications.

Response to REVIEWERS' COMMENTS

Reviewer 3 (Remarks to the Author): I believe the response is sufficient to address the comments of Reviewer 4. It is a long-standing issue whether and how near-interface order affects the interfacial free energy. The authors made a good argument in this paper that interfacial ordering is general while dendrite orientation transition is not. They attributed the key factor to the mismatch of Sm and Al layering in the current case. After the long review, the current manuscript presents a consistent combination of experiments, phase field and atomic simulations. In my opinion, the current manuscript is acceptable for publication in Nature Communications.

We thank reviewer 3 for the positive opinion on our response.